# A New Method for Inducing Mental Fatigue: A High Mental Workload Task Paradigm Based on Complex Cognitive Abilities and Time Pressure

**DOI:** 10.3390/brainsci15060541

**Published:** 2025-05-22

**Authors:** Lei Ren, Lin Wu, Tingwei Feng, Xufeng Liu

**Affiliations:** 1Military Psychology Section, Logistics University of PAP, Tianjin 300309, China; rl_fmmu@163.com; 2Military Mental Health Services & Research Center, Tianjin 300309, China; 3Department of Military Medical Psychology, The Fourth Military Medical University, Xi’an 710032, Chinaftw_fmmu@163.com (T.F.)

**Keywords:** mental fatigue, high mental workload, complex cognitive abilities, time pressure, time on task

## Abstract

**Objectives**: With the advancement of modern society, people in cognitively demanding jobs are increasingly exposed to occupational stress. Prolonged and high-intensity cognitive activities are prone to inducing mental fatigue (MF), which adversely affects both psychological and physiological well-being, as well as task performance. Existing methods for inducing MF often demonstrate limited effectiveness due to insufficient cognitive load from overly simplistic tasks and the potential emotional disturbance caused by prolonged task duration. This study aims to explore a comprehensive cognitive task paradigm that integrates task complexity and time pressure, thereby developing a novel and effective method for inducing MF based on high mental workload (HMW) and the effects of time on task (ToT). **Methods**: Using convenience sampling, university students from a medical college were recruited as participants. The study was conducted in three steps. In the first step, we constructed a 1-back Stroop (BS) task paradigm by designing tasks with varying levels of complexity and incorporating time pressure through experimental manipulation. In the second step, the efficacy of the BS task paradigm was validated by comparing it with the traditional 2-back cognitive task in inducing HMW. In the third step, an MF induction protocol was established by combining the BS task paradigm with the ToT effect (i.e., a continuous 30 min task). Effectiveness was assessed using validated subjective measures (NASA Task Load Index [NASA-TLX] and Visual Analog Scale [VAS]) and objective behavioral metrics (reaction time and accuracy). Statistical analyses were performed using analysis of variance (ANOVA) and *t*-tests. **Results**: The BS task paradigm, which integrates complex cognitive abilities such as attention, working memory, inhibitory control, cognitive flexibility, and time pressure, demonstrated significantly higher NASA-TLX total scores, as well as elevated scores in mental demand, temporal demand, performance, and frustration scales, compared to the 2-back task. Additionally, the BS task paradigm resulted in longer reaction times and lower accuracy. As the BS task progressed, participants exhibited significant increases in mental fatigue (MF), mental effort (ME), mental stress (MS), and subjective feelings of fatigue, while the overall number of correct trials and accuracy showed a significant decline. Furthermore, reaction times in the psychomotor vigilance test (PVT) were significantly prolonged, and the number of lapses significantly increased between pre- and post-task assessments. **Conclusions**: The BS task paradigm based on complex cognitive abilities and time pressure could effectively induce an HMW state. Combined with the ToT effect, the BS paradigm demonstrated effective MF induction capabilities. This study provides a novel and reliable method for inducing HMW and MF, offering a valuable tool for future research in related fields.

## 1. Introduction

### 1.1. MF and Theoretical Models

MF refers to a psychophysiological state induced by sustained cognitive activity, characterized primarily by increased subjective fatigue and diminished cognitive performance [1,2,3,4]. In fact, many human activities, such as learning, collaboration, traffic participation, and competitive sports, place high cognitive demands on individuals, often leading to MF when such activities are prolonged [5,6]. Despite over a century of research, the principles and mechanisms underlying MF remain elusive, with no unified theory established [2,7]. To date, researchers have proposed various models to explain MF, including motivational control theory [8,9,10], underload theory [11,12], resource depletion theory [13,14], and waste disposal theory [5,15]. Among these, resource depletion theory has been widely accepted, suggesting that MF results from the sustained allocation of cognitive resources to tasks, leading to depletion of resources that cannot be readily replenished [13,14]. Due to the varying degrees of cognitive control required in cognitive tasks, researchers believe that depletion of cognitive resources typically refers to depletion in sustained tasks [2,6]. With the development of cognitive neuroscience, researchers have started to explore the neural mechanisms of MF in addition to explaining its occurrence at the theoretical level. Two important meta-analyses on the neural mechanisms of time-on-task (ToT) effects show that mental fatigue from ToT is linked to reduced activity in mainly right-lateralized brain regions (e.g., dorsolateral prefrontal cortex and anterior cingulate cortex) [2,12]. Salihu et al. pointed out that as MF progresses, the activation of the brain’s cognitive control network decreases, while the activation of some areas of the default mode network increases [2]. Also, Qi et al. summarized and reviewed the brain functional connectivity studies on MF and found that under MF conditions the brain’s global integration generally declines, while local specificity increases [7]. It should be noted that despite the numerous cognitive neuroscience-related studies on MF, fully understanding the neural mechanisms behind MF remains challenging [2,7].

### 1.2. Methods for Inducing MF Through Cognitive Tasks

Generally, researchers use laboratory cognitive tasks [1,2,3,16,17], simulation tasks (e.g., simulated driving or flying) [18,19,20,21], or sleep deprivation tasks [22,23,24,25] to induce MF. Traditional laboratory cognitive tasks are preferred due to their simplicity, low cost, high controllability, and repeatability [16,17]. Common tasks for inducing MF include the Stroop task [26,27,28], n-back task [29,30,31], and PVT [32,33,34]. In addition, in order to enhance ecological validity and distinguish MF from boredom caused by repetitive simple tasks, researchers have developed some complex laboratory tasks, such as the time load dual back (TloadDback) task [35,36,37], multi-attribute task battery-II (MATB-II) [38], Toulouse n-back task (TNT) [39], and air traffic control (ATC) task [40].

Previous studies have shown that the ability of laboratory tasks to induce MF depends on two primary characteristics: task difficulty and duration [1]. On one hand, the ability to induce MF is influenced by task duration—the longer the task duration, the higher the likelihood of MF occurrence. In fact, most studies manipulate task duration to induce MF, referred to as the ToT effect [35,41,42]. The ToT approach has achieved considerable success in MF induction. Researchers also note that prolonged tasks, regardless of difficulty, eventually lead to MF, albeit at varying rates [35,41]. In general, tasks of insufficient duration may result in low MF levels, while excessively long tasks can cause boredom, frustration, or withdrawal of effort [43]. On the other hand, task difficulty represents the cognitive demands of the task; higher difficulty increases cognitive resource consumption, leading to greater MF [13,14]. Recent studies by O’Keeffe et al. compared the MF induction efficacy of five different tasks, demonstrating that a relatively difficult individual TloadDback task induced greater MF in a shorter duration (16 min) task compared to the AX-continuous performance test (AX-CPT) task sustained for 90 min [44]. This finding underscores the temporal importance of tasks in MF induction.

Mental workload (MWL) refers to the outcome of an individual’s limited cognitive processing capacity in response to task demands [45,46,47,48]. Consequently, higher task demands are associated with increased MWL. Previous studies suggest that task demands generally encompass task complexity and time pressure [47]. In contemporary society, many professionals (e.g., competitive athletes, military personnel, commercial pilots, and drivers) perform tasks that combine complexity and time pressure, requiring sustained high-level cognitive control and attentional abilities while responding rapidly to unpredictable situations. Such sustained HMW conditions can lead to MF [5,49,50,51], providing a unique perspective for effectively inducing MF. Inducing MF via HMW conditions avoids the low induction levels caused by simplistic tasks [44] and mitigates the boredom and frustration associated with prolonged task durations [2,11,52]. Compared to traditional laboratory tasks, HMW task paradigms offer higher efficiency and “purity” in MF induction [1]. Thus, an HMW task paradigm incorporating task complexity and time pressure presents both theoretical and practical advantages in MF induction.

This study developed an HMW cognitive task paradigm (i.e., the BS paradigm) by increasing task demands through task complexity and time pressure. Subsequently, a new MF induction method was developed, which applied the ToT approach to the BS paradigm, providing an innovative approach for effectively inducing HMW conditions and MF.

## 2. Methods

### 2.1. Research Design

The experimental process consisted of three steps:

(1) Task Complexity and Time Pressure:

As mentioned in the introduction, task demands generally include task complexity and time pressure. Therefore, this study will build an HMW task paradigm by increasing task demands in terms of task complexity and time pressure.

Construction of task complexity for BS cognitive task paradigm: The BS cognitive task paradigm was constructed by combining the spatial 1-back task (a kind of working memory task) with the color–word Stroop task (a kind of inhibitory control task) (see Section 2.3.1). In the BS cognitive task paradigm, participants are required to use four cognitive abilities: attention (focused and directed attention for each trial), working memory (storing, maintaining, and updating spatial location information in the 1-back task), inhibitory control (inhibiting the dominant response to congruent color–word combinations in the Stroop task), and cognitive flexibility (switching between the spatial 1-back task and Stroop task). The complexity of the task is relatively high.

Construction of time pressure for BS cognitive task paradigm: The BS cognitive task paradigm consisted of 121 trials (the first trial did not require a key press, then from the second trial onward the four conditions appeared pseudo-randomly 30 times each). Additionally, participants’ key responses were counterbalanced (i.e., some participants pressed the “F” key for conditions 1 or 2, and the “J” key for conditions 3 or 4, while others did the opposite). Participants were instructed that there was no time limit for responding to each trial and that the next trial would appear immediately after the key press. They were also instructed to respond as quickly and accurately as possible. Since there was no time limit for trials in the BS cognitive task paradigm, the inter-trial interval included two types of feedback: “√” (correct) and “×” (incorrect), displayed in Courier New font size 36. Before the official experiment, participants first read the instructions themselves and then went through a practice phase (16 trials per block). Participants could only proceed to the official experiment phase after achieving 90% or higher accuracy during the practice phase. Siegel and Wolf noted that time pressure could be controlled by adjusting the ratio between the required time (RT) and the available time (AT) in the task [53]. The RT can be determined without any response time limits in the task [54,55]. Based on previous recommendations, the average RT under no-response-time-limited conditions was used as the RT for the task [54,56,57]. Furthermore, it has been found that when the RT to AT ratio is 0.8, individuals experience a sense of time pressure [54]. Therefore, the AT for the BS cognitive task paradigm in this study was set as 1.25 times the average RT under no-response-time-limited conditions.

(2) Validation of HMW induction efficacy: To compare the constructed BS cognitive task paradigm with the 2-back task to validate its efficacy in inducing HMW, participants were required to complete two tasks, namely the BS cognitive task and the 2-back task. To minimize the influence of confounding factors such as fatigue and practice, two experimental controls were conducted. First, the order of the two tasks was balanced within the subjects, with at least a 3 min break in between. Second, both tasks were block designs. Specifically, both the BS task and the 2-back task consisted of 8 blocks, each with 8 trials that required the subject to press a button (4 conditions were presented pseudo-random twice in each block). The rest time between blocks lasted for 20 s. After completing each task, participants were required to answer the NASA-TLX. The experimental flow of the second step is shown in Figure 1.

(3) Validation of MF Induction Efficacy: Combining the BS task paradigm with the ToT effect to construct the method in MF induction and evaluate its efficacy. Participants were required to complete a 30 min BS cognitive task consisting of 6 blocks. Within each block, the four conditions were presented in a pseudo-random manner of 1:1:1:1. After each block, VAS (an effective tool for subjective measurement of MF, consisting of four questions: MF, ME, MS, and boredom) evaluation was conducted. Before the first block, VAS was used to assess the baseline level of MF. In addition, participants were also required to complete a 3 min PVT test before and after the task. The experimental flow of the third step is shown in Figure 2. Before the formal experiment, participants were required to first practice and enter the formal experimental stage only when their accuracy reached 90% or above. After the practice session, participants had a 5–10 min break to eliminate potential MF caused by the practice.

### 2.2. Participants

Participants were recruited through posters at The Fourth Military Medical University. Inclusion criteria: (1) male; (2) aged 18–23 years old; (3) normal vision or corrected to normal. Exclusion criteria: (1) history of psychiatric or neurological disorders; (2) use of medications that may affect brain or autonomic nervous system function; (3) color blindness or color weakness; (4) left-handedness. In the first step of the study, a total of 100 male military academy cadets were recruited. All participants completed the experiment, and none dropped out. The final sample included 94 participants (6 participants were excluded due to poor behavioral data quality). The average age of the remaining participants was 20.05 ± 1.15 years old, with an average education of 14.98 ± 0.98 years (years of formal education, the same below). In the second step of the study, 36 male cadets were enrolled, and all participants completed the experiment. The average age was 19.78 ± 0.72 years old, and the average education was 14.67 ± 0.76 years. In the third step of the study, 72 male cadets were enrolled, and all participants completed the experiment. The average age was 20.01 ± 0.62 years old, and the average education was 14.99 ± 0.21 years. There was no overlap of participants across the three steps of the study. All participants signed written informed consent. The study adhered to the principles of the Helsinki Declaration and was approved by the Ethics Committee of the First Affiliated Hospital of The Fourth Military Medical University (KY20242053-C-1).

### 2.3. Materials

#### 2.3.1. BS Task Paradigm

The BS task paradigm was used in the second and third steps of the study, and the flow of the BS task paradigm is shown in Figure 3. The BS cognitive task paradigm includes four conditions and two types of key responses. During the task, participants were required to judge whether the position of a Chinese character in the current trial matched the position in the previous trial (equivalent to a spatial 1-back task with a 3 × 3 grid of nine squares) and whether the color and meaning of the Chinese character in the current trial were consistent (equivalent to a color–word Stroop task with four colors: red, yellow, blue, green, and corresponding meanings inside the square). Key responses were required based on these judgments. In each trial, the spatial 1-back task and the color–word Stroop task had 50% consistency and 50% inconsistency conditions, respectively. Each trial could meet one of the following four conditions. Condition 1: both spatial 1-back and Stroop tasks are consistent; Condition 2: both spatial 1-back and Stroop tasks are inconsistent; Condition 3: spatial 1-back is consistent and Stroop is inconsistent; Condition 4: spatial 1-back is inconsistent and Stroop is consistent. The key responses were as follows: press the “F” key when the trial met Conditions 1 or 2; press the “J” key when the trial met Conditions 3 or 4. The time pressures for Conditions 1, 2, 3, and 4 were 2300 ms, 3000 ms, 2600 ms, and 2800 ms, respectively (the construction process of time pressure is described in Section 2.1). Time pressure was displayed as a green progress bar at the top of each trial, with indentations at both ends. The interstimulus interval (ISI) was the feedback to participants’ key responses, lasting 500 milliseconds, including “√” (correct), “×” (incorrect), and “–” (no response). After a key press, the next trial would immediately appear.

#### 2.3.2. 2-Back Task

The 2-back task was used in the second step of the study. In the 2-back task, 26 letters of the English alphabet (in Courier New font, size 36) were presented pseudo-randomly, and participants were required to judge whether the current trial’s letter matched the letter from two trials ago, with a 50% match rate. Consistent with previous studies, each letter was displayed for 0.5 s, followed by a 2 s blank screen [58,59,60,61]. No feedback was provided during the task. Participants were instructed to respond as quickly and accurately as possible.

#### 2.3.3. NASA-TLX

The NASA-TLX was used in the second step of the study to assess MWL. The NASA-TLX is a widely used and effective tool for evaluating MWL, with six dimensions: mental demand, physical demand, time demand, performance, effort, and frustration [62,63,64]. Scores on each dimension range from 0 (very low) to 20 (very high). In this study, we used the unweighted total score and scores on each dimension to evaluate the overall and specific dimensions of MWL induced by the BS cognitive tasks [62,65].

#### 2.3.4. PVT

The PVT was used in the third step of the study. PVT is a widely used simple reaction time task that measures participants’ alertness, arousal levels, and attention. It has the advantages of simple operation, small practice effects, low influence of individual differences, high reliability, and sensitivity, and has been widely used in the field of fatigue assessment [66,67]. In the PVT, participants must react quickly to a target stimulus presented at random intervals (ranging from 2 to 10 s, including 1 s to read the response time after each key press). The target stimulus is a red circle with a diameter of 10 cm. A response time of 100 ms or more is considered a valid response, and a response time greater than or equal to 500 ms is considered an attention lapse [68]. The PVT evaluation metrics in this study were average reaction time and number of attention lapses [68,69,70,71].

#### 2.3.5. VAS

The VAS was used in the third step of the study for subjective assessment. The VAS is a simple, commonly used, and effective method for measuring MF [1,4,72,73]. In this study, VAS included four items assessing MF, ME, and boredom. Participants were instructed to slide a marker along a line labeled “none at all” at one end and “very much” at the other end, to their perceived level of the given items, and then click the “submit” button to proceed to the next question. The final score was recorded in the software backend but was not shown to participants, with a range of 0 to 100 and a minimum unit of 1. Participants were informed that they could not rest during the VAS assessment, and after completing VAS, they would immediately proceed to the next task block.

### 2.4. Statistical Analysis

#### 2.4.1. Statistical Analysis for Construction of Task Complexity and Time Pressure for BS Cognitive Task Paradigm

Due to the presence of extreme values in RTs under no-response-time-limited conditions, data preprocessing was conducted. First, incorrect trials were removed for each participant. Then, outliers were removed for each participant within each condition (if the reaction time for a single trial was greater than the mean of that trial’s condition plus 3 standard deviations, or smaller than the mean minus 3 standard deviations, i.e., >Mean + 3SD or <Mean − 3SD) [74]. Finally, outliers at the group level were removed (if the mean for a given condition in a participant was greater than the group mean for that condition plus 3 standard deviations, or smaller than the group mean minus 3 standard deviations, i.e., >Mean + 3SD or <Mean − 3SD) [74]. After preprocessing, data from 6 participants were excluded, leaving 94 participants’ data for analysis. Since accuracy was controlled during the study (i.e., participants could only proceed to the official phase after reaching 90% or higher accuracy in the practice phase), and the time pressure was mainly related to reaction time, statistical analysis was conducted on the reaction times for each condition. Data were analyzed using SPSS 25. Descriptive statistics were first calculated for the reaction times and accuracy in each condition. Then, after conducting normality tests (Kolmogorov–Smirnov) on the reaction times for each condition, one-way repeated measures ANOVAs were performed, followed by post-hoc comparisons using Bonferroni corrections, to explore the impact of different conditions on reaction times and whether there were statistically significant differences in reaction times between pairs of conditions.

#### 2.4.2. Statistical Analysis for HMW Induction Validity

Data analysis for the second step was performed using SPSS 25. First, descriptive statistics were calculated for the subjective scales and objective behavioral results of the two tasks. Next, normality tests (Shapiro–Wilk) were performed on the differences in the subjective scales and objective behavioral results, followed by matched *t*-tests (for normally distributed data) or Wilcoxon signed-rank tests (for non-normally distributed data).

#### 2.4.3. Statistical Analysis for MF Induction Validity

Data analysis for the third step was performed using SPSS 25. First, descriptive statistics were calculated for the subjective scales (i.e., MF, ME, MS, and boredom) and objective behavioral data (i.e., correct trial counts and accuracy) within different blocks. Next, normality tests (Kolmogorov–Smirnov) were performed on the data, followed by one-way repeated measures ANOVAs (for normally distributed data) or Friedman tests (for non-normally distributed data) with post hoc comparisons (using Bonferroni corrections for multiple comparisons). Descriptive statistics were also performed for pre- and post-test PVT behavioral data. Finally, matched sample differences were analyzed using matched *t*-tests (for normally distributed data) or Wilcoxon signed-rank tests (for non-normally distributed data). Additionally, the relationships between subjective and behavioral indicators within each block were calculated using Spearman correlations.

## 3. Results

### 3.1. Construction of Task Complexity and Time Pressure for BS Cognitive Task Paradigm

The average reaction times and accuracy results for the four conditions of the BS cognitive task paradigm under no-response-time-limited conditions are shown in Table 1.

Normality tests on the reaction times for the BS cognitive task paradigm under no-response-time-limited conditions indicated that reaction times for all four conditions satisfied the normality assumption (*ps* > 0.05). Therefore, a one-way repeated measures ANOVA was used to explore the effect of different conditions on reaction time. Mauchly’s sphericity hypothesis test indicated that the variance–covariance matrix for the dependent variables was unequal (χ^2^_(5)_ = 14.13, *p* < 0.05) and the Greenhouse–Geisser correction was applied (ε = 0.90). The final results showed that the effect of condition on reaction times was statistically significant (F_(2.71, 6,361,503.39)_ = 74.84, *p* < 0.001, partial η^2^ = 0.45). Post-hoc comparisons revealed that there were significant differences in the mean reaction times between any two conditions (*ps* < 0.002). In order to ensure that participants experienced approximately the same sense of time pressure while responding to the four different conditions, we calculated the AT for each condition. As mentioned earlier, AT was set to 1.25 times the average RT. Therefore, the AT for conditions 1, 2, 3, and 4 were 2300 ms, 3000 ms, 2600 ms, and 2800 ms, respectively. After adjusting for time pressure, the final version of the BS cognitive task paradigm was established, as illustrated in Figure 3 of the manuscript.

### 3.2. Validation of HMW Induction Effectiveness

#### 3.2.1. Comparison of Subjective Scale Indicators

The results of the tests for differences in subjective scale indicators are presented in Table 2 and Figure 4.

#### 3.2.2. Comparison of Behavioral Indicators

The results of the statistical tests for differences in behavioral indicators (reaction time and accuracy) are shown in Table 2 and Figure 5.

### 3.3. Validation of MF Induction Effectiveness

#### 3.3.1. Comparison of VAS Indicators

The results for MF showed that time had significant influence (χ^2^_(6)_ = 310.89, *p* < 0.001). For ME, the effect of time was statistically significant (χ^2^_(5)_ = 58.45, *p* < 0.001). For MS, time significantly influenced the result (χ^2^_(5)_ = 43.71, *p* < 0.001). Boredom was also significantly affected by time (χ^2^_(5)_ = 72.97, *p* < 0.001). The results are presented in Table 3 and Figure 6.

#### 3.3.2. Comparison of BS Cognitive Task Behavioral Indicators

The behavioral indicators included correct trials and accuracy of BS cognitive task results. Time significantly affected the number of correct trials (F_(2.68, 2966.19)_ = 17.43, *p* < 0.001, partial η^2^ = 0.20). For accuracy, time significantly influenced the result (χ^2^_(5)_ = 66.91, *p* < 0.001). Additionally, the decline in the number of correct trials between Block 1 and Block 6 was 6.25% and the decline in accuracy was 3.07%. The results are presented in Table 3 and Figure 7.

#### 3.3.3. Spearman Correlations Between Subjective and Behavioral Indicators

In the Spearman correlation matrix of subjective and behavioral indicators, stronger significant correlations were observed within their respective categories. For specific correlation values, please refer to Figure 8.

#### 3.3.4. Comparison of PVT Task Behavioral Indicators

The behavioral indicators included reaction time and attention lapses on the PVT task. The results showed that reaction time in the post-test was significantly longer than in the pre-test (z_(71)_ = −5.03, *p* < 0.001). The number of attention lapses in the post-test was also significantly higher than that in the pre-test (z_(71)_ = −3.28, *p* = 0.001). The results are presented in Table 4 and Figure 9.

## 4. Discussion

### 4.1. Validation of HMW Induction Effectiveness

In the BS cognitive task paradigm for inducing HMW, the results showed that the BS cognitive task induced significantly higher subjective MWL (as measured by the total NASA-TLX score) than the 2-back task. This indicates that the BS task paradigm is an effective laboratory cognitive task for inducing subjective HMW. Among the six dimensions of the NASA-TLX, the BS task paradigm had significantly higher scores in the mental demand, time demand, performance, and frustration dimensions compared to the 2-back task. This result is consistent with expectations. In terms of task complexity, the 2-back task only requires participants to judge whether the current trial matches the trial two steps prior (working memory), while the BS cognitive task requires participants to judge whether the positions of Chinese characters in the current and previous trials match (working memory), whether the color and meaning of the characters in the current trial match (inhibitory control), and to determine which condition the combination of these two meets. Additionally, as for the BS task paradigm, we specifically set time pressure to induce a sense of urgency in participants. Time pressure, as a form of stress, can elicit negative emotions in individuals [75,76]. Furthermore, more conflicts and errors during the task execution could lead to greater negative emotions [4,77]. Compared to the 2-back task, the BS task paradigm involves more conflicts (due to its higher complexity and time pressure) and a higher error rate. Behavioral results showed that the BS task paradigm had significantly higher reaction time and lower accuracy compared to the 2-back task. This is reasonable because the BS task paradigm involves more cognitive abilities, requires more judgmental steps, and has greater time pressure, making it expected that the reaction time would be longer and accuracy lower compared to the 2-back task.

### 4.2. Validation of MF Induction Effectiveness

In subjective scales, MF increased rapidly as the task progressed, and statistical analysis showed that time had a significant effect on MF. In fact, in many different types of sustained cognitive tasks, subjective MF has been shown to increase with task duration [35,44,73,78]. Compared to previous studies using the PVT, AX-CPT, and Stroop tasks, the BS task in this study had a shorter duration (only 30 min) but induced a higher level of subjective MF, which suggests that the BS cognitive task is more efficient in inducing MF. It is worth noting that the increase in subjective MF reduces gradually with the progress of the task time, with the first block increasing the most, followed by the second block, and then gradually less increase. This phenomenon is similar to diminishing marginal effects and may indicate that subjective MF has certain nonlinear characteristics, which can be further explored in future research. On the subjective scale, ME gradually increased and remained relatively high throughout the task. Generally, individuals rely on non-automated cognitive control processes for difficult tasks, leading to increased ME [79]. Previous studies suggest that ME has high cost and aversion characteristics and is maintained or increased when individuals expect benefits from activities [79]. Therefore, participants likely maintained high motivation throughout the sustained task, which reflects the higher difficulty level of the BS task paradigm. MS in the subjective scale generally increased over time, consistent with findings from a 40 min vigilance task that showed increasing stress over time [80]. Boredom gradually increased on the subjective scale but remained at a relatively low level. Boredom is an emotional state in which it is difficult to participate in and pay attention to internal or external information, characterized by a lack of core motivation and disengagement from the task [81,82]. According to the underload theory of MF, some cognitive tasks that induce MF (such as the Stroop and PVT tasks) are relatively simple and monotonous, which may lead to increased boredom over time [52,83], causing reduced task engagement and increased mind wandering. These may lead to poor performance [7,11]. In contrast, the BS task paradigm had higher cognitive demands, requiring sustained effort (high task engagement) and resulting in fewer declines in performance due to off-task thinking.

The behavioral indicators of the BS task paradigm showed that the number of correct trials and accuracy gradually declined as the task progressed, and statistical analysis indicated that time significantly influenced these indicators. Additionally, the behavioral results of the PVT showed that the post-test reaction time and the number of attention lapses were significantly higher than in the pre-test, with statistically significant differences. The PVT has high reliability and sensitivity and has been widely used in the field of MF assessment [68,69,70,71]. Some researchers consider the decline in behavioral performance during the task to be the “gold standard” for measuring MF [73,84]. Therefore, the behavioral results (including those from the BS task paradigm and PVT) confirm that this study successfully induced MF. As mentioned earlier, regardless of task difficulty, prolonged tasks ultimately lead to MF in individuals, although the rates at which MF develops may vary [35,41]. In many previous studies, researchers used relatively simple tasks of longer duration to induce MF. This approach often results in declines in MF-related behavioral performance, with debates over whether such declines are due to MF or boredom (underload theory) [2,11,52]. Some studies suggest that MF induced by relatively simple, long-duration tasks is less pronounced [44,85]. Moreover, prior research has even found that this MF induction method failed to cause a decline in MF-related behavioral performance. For example, Smith et al. did not observe a decline in task-related behavioral indicators during a 45 min AX-CPT and Stroop task [73], and Verschueren et al. [86] and Noé et al. [85] also failed to observe a decline in task-related performance during 90 min Stroop and AX-CPT tasks. In contrast to these studies, the BS task paradigm induced a significant decline in MF-related behavioral indicators after just 15 min approximately. Therefore, this study may provide a more efficient cognitive task model for the MF research field.

The correlation results showed that stronger significant correlations appeared within the subjective and behavioral indicators individually, suggesting the dissociation of subjective and objective indicators of MF [87,88]. Many studies have found no correlations between subjective and objective measurements of MF [4,72,89,90,91]. Researchers think this may be due to the poor reliability of behavioral tests and the different response processes involved in the measurement of different domains [72,92,93]. Note that in the first three blocks, there was a negative correlation between count and mental stress, indicating that mental stress may affect behavioral performance. This is supported by previous studies showing that stress can impair cognitive function and behavioral performance [94,95,96,97].

Several limitations of this study should also be acknowledged. First, the exclusive recruitment of male military cadets resulted in a highly homogeneous sample, which may limit the ecological validity of the findings for populations with different demographics or occupational backgrounds. Second, potential confounding factors including inter-individual differences in cognitive capacity, arousal levels, and task motivation could have influenced task performance outcomes. Third, in the process of time pressure setup, previous studies have varied in their understanding of task RT [98]. Therefore, whether the AT set in the current study is the “optimal solution” remains to be explored and verified. Fourth, the validation of the BS task paradigm relied solely on behavioral indicators and subjective ratings, without employing neurophysiological measures (e.g., electroencephalogram, cerebral blood oxygenation, heart rate variability, eye-tracking features), potentially limiting mechanistic insights into HMW and MF. In future studies, we will recruit more representative and diversified samples with larger sample sizes, while employing multimodal assessment approaches to enable more comprehensive investigations of HMW and MF.

## 5. Conclusions

This study developed a new method for inducing MF based on the BS task paradigm (a kind of HMW task paradigm), which integrates complex cognitive abilities (attention, working memory, inhibitory control, and cognitive flexibility) and time pressure, along with the ToT effect. Through self-report scales and behavioral experiments, the BS task paradigm based on complex cognitive abilities and time pressure was shown to effectively induce an HMW state. The BS task paradigm, combined with ToT, has good MF-inducing effects and avoids the shortcomings of existing laboratory methods, which may fail to induce sufficient MF due to the simplicity of tasks or result in emotional issues from excessive task duration. This study provides a new method for inducing HMW and MF, which is of great significance for laboratory research and intervention applications of HMW and MF and can be widely applied in related studies in the future.

## Figures and Tables

**Figure 1 brainsci-15-00541-f001:**
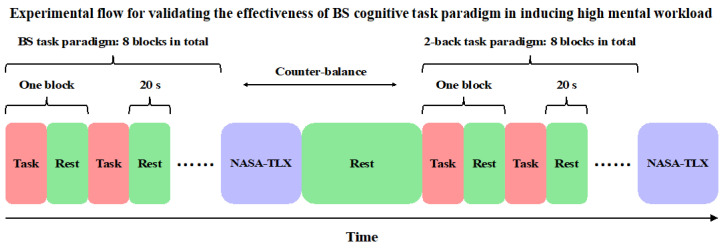
Experimental flow for validating the effectiveness of the BS cognitive task paradigm in inducing high mental workload.

**Figure 2 brainsci-15-00541-f002:**
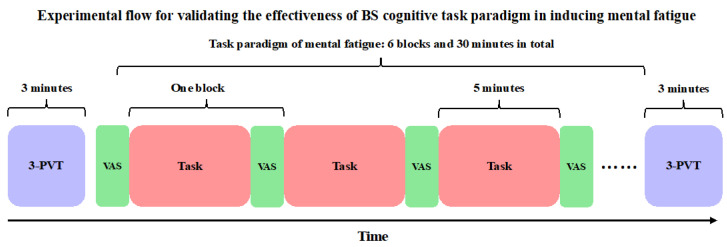
Experimental flow for validating the effectiveness of the BS cognitive task paradigm in inducing mental fatigue.

**Figure 3 brainsci-15-00541-f003:**
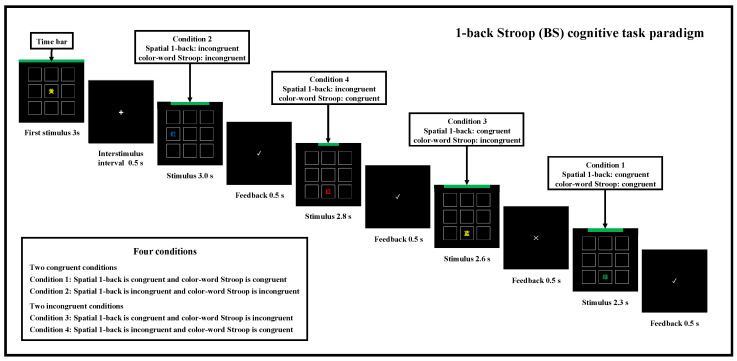
1-back Stroop (BS) cognitive task paradigm. Note: 红 (Chinese) = Red (English); 黄 (Chinese) = Yellow (English); 蓝 (Chinese) = Blue (English); 绿 (Chinese) = Green (English).

**Figure 4 brainsci-15-00541-f004:**
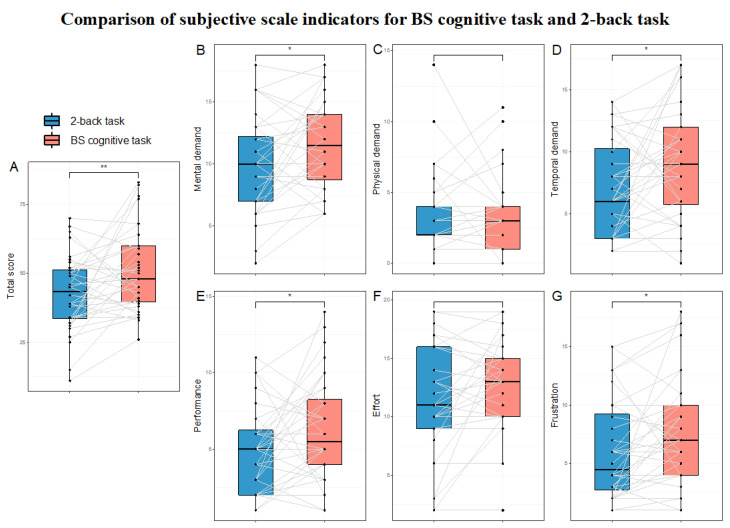
Comparison of subjective scale indicators for BS cognitive task and 2-back task. Note: Black dots represent the participant data, and the gray-white lines connect the data of the same participant across the two tasks (n = 36). In the box plots, the upper and lower lines of the box represent the upper and lower quartiles, and the middle line represents the median. (**A**) NASA-TLX total score. (**B**) Mental demand dimension. (**C**) Physical demand dimension. (**D**) Time demand dimension. (**E**) Performance dimension. (**F**) Effort dimension. (**G**) Frustration dimension. NASA-TLX—NASA Task Load Index. * *p* < 0.05, ** *p* < 0.01.

**Figure 5 brainsci-15-00541-f005:**
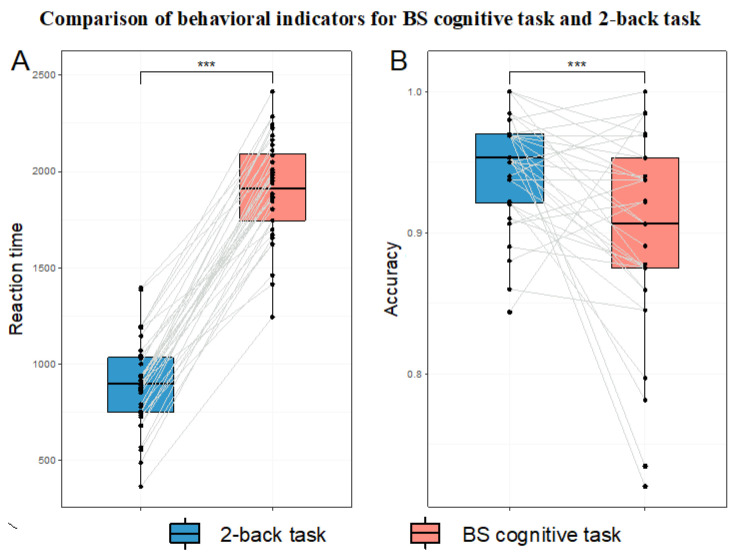
Comparison of behavioral indicators for BS cognitive task and 2-back task. Note: Black dots represent participant data, and the gray-white lines connect the data of the same participant across the two tasks (n = 36). In the box plots, the upper and lower lines of the boxes represent the upper and lower quartiles and the middle line represents the median. (**A**) Reaction time. (**B**) Accuracy. *** *p* < 0.001.

**Figure 6 brainsci-15-00541-f006:**
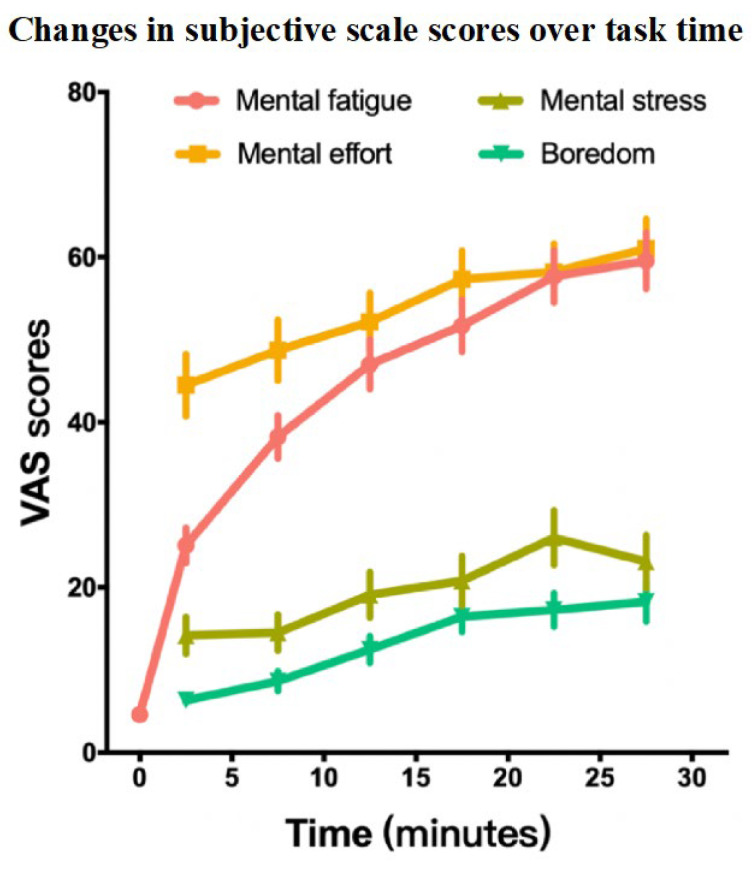
Changes in subjective scale scores over task time. Note: The error bar is the standard error. VAS—Visual Analog Scale.

**Figure 7 brainsci-15-00541-f007:**
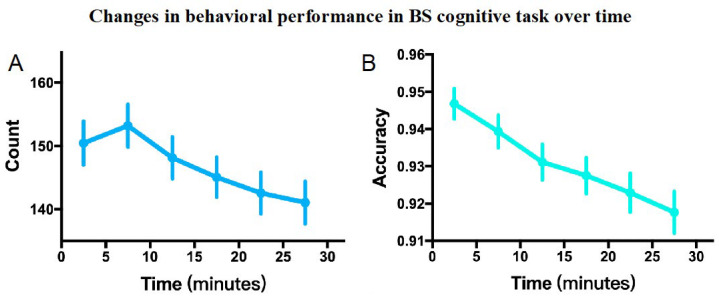
Changes in behavioral performance in BS cognitive task over time. Note: The error bar is the standard error. (**A**) Correct number of trials. (**B**) Accuracy rate.

**Figure 8 brainsci-15-00541-f008:**
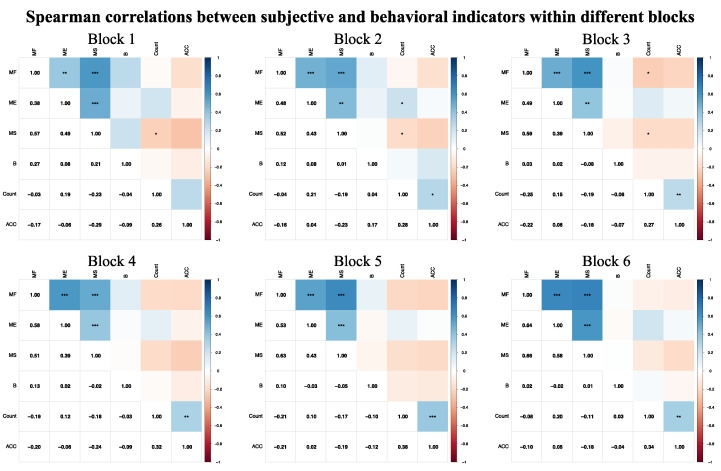
Spearman correlations between subjective and behavioral indicators within different blocks. Note: MF—mental fatigue; ME—mental effort; MS—mental stress; B—boredom; ACC—accuracy. * *p* < 0.05, ** *p* < 0.01, *** *p* < 0.001.

**Figure 9 brainsci-15-00541-f009:**
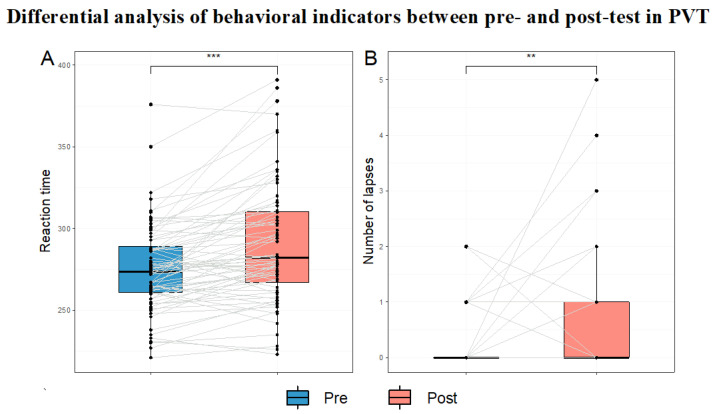
Differential analysis of behavioral indicators between pre- and post-test in PVT. Note: Black dots represent the participant data, and the gray-white lines connect the data of the same participant across the PVT task (n = 72). In the box plots, the upper and lower lines of the boxes represent the upper and lower quartiles and the middle line represents the median. (**A**) Reaction time. (**B**) Attention lapse count. ** *p* < 0.01, *** *p* < 0.001.

**Table 1 brainsci-15-00541-t001:** Behavioral results of BS cognitive task paradigm under no-response-time-limited conditions.

Variables	Mean	SD
Reaction Times (ms)		
Condition 1	1815.63	503.94
Condition 2	2394.17	572.98
Condition 3	2110.47	558.54
Condition 4	2251.46	545.22
Accuracy		
Condition 1	0.92	0.07
Condition 2	0.88	0.09
Condition 3	0.91	0.06
Condition 4	0.91	0.07

**Table 2 brainsci-15-00541-t002:** Comparison results of scales and behavioral indicators in the two tasks.

Indicator	BS Cognitive Task		2-Back Task		*p*	Effect Size
	Mean	SD	Mean	SD		
Subjective Scale						
NASA-TLX	50.67	14.80	42.56	13.69	0.008 ^a^	0.472 ^c^
Mental demand	11.64	3.51	9.94	3.83	0.014 ^a^	0.429 ^c^
Physical demand	3.08	2.62	3.22	2.80	0.781 ^b^	0.069 ^d^
Time demand	9.11	4.67	7.03	3.86	0.024 ^a^	0.394 ^c^
Performance	6.31	3.34	4.89	2.90	0.029 ^a^	0.381 ^c^
Effort	12.81	3.71	11.42	4.73	0.110 ^b^	0.340 ^d^
Frustration	7.72	4.56	6.06	4.23	0.015 ^b^	0.510 ^d^
Behavioral Results						
Reaction Time (ms)	1910	260	902	232	<0.001 ^a^	4.321 ^c^
Accuracy (%)	0.90	0.07	0.95	0.04	<0.001 ^b^	−0.671 ^d^

Note: ^a^ Matched *t*-test; ^b^ Wilcoxon signed-rank test; ^c^ Cohen’s d; ^d^ rank-biserial correlation; NASA-TLX: NASA Task Load Index.

**Table 3 brainsci-15-00541-t003:** Descriptive statistics and difference test results of indicators within different blocks.

Indicator	Pre-Test	Block 1	Block 2	Block 3	Block 4	Block 5	Block 6	F/χ^2^	*p*	Effect Size
	Mean ± SD	Mean ± SD	Mean ± SD	Mean ± SD	Mean ± SD	Mean ± SD	Mean ± SD			
Subjective Scale										
MF	4.60 ± 5.35	25.08 ± 18.13	38.24 ± 22.23	47.00 ± 25.38	51.71 ± 27.09	57.65 ± 27.07	59.56 ± 28.89	310.89	<0.001 ^b^	0.720 ^d^
ME	—	44.51 ± 32.05	48.75 ± 31.17	52.17 ± 29.72	57.32 ± 29.38	58.21 ± 29.16	61.03 ± 30.65	58.45	<0.001 ^b^	0.162 ^d^
MS	—	14.18 ± 18.84	14.51 ± 18.60	19.10 ± 23.55	20.81 ± 25.46	26.01 ± 28.13	23.11 ± 27.45	43.71	<0.001 ^b^	0.121 ^d^
Boredom	—	6.33 ± 8.14	8.64 ± 10.25	12.50 ± 13.97	16.49 ± 15.78	17.28 ± 17.31	18.26 ± 20.32	72.97	<0.001 ^b^	0.203 ^d^
Behavioral Results										
Correct trials	—	150.47 ± 29.27	153.22 ± 28.69	148.14 ± 27.91	145.06 ± 27.00	142.56 ± 27.99	141.07 ± 28.72	17.43	<0.001 ^a^	0.200 ^c^
Accuracy (%)	—	94.68 ± 3.50	93.94 ± 3.72	93.11 ± 4.05	92.75 ± 4.11	92.29 ± 4.42	91.77 ± 4.72	66.91	<0.001 ^b^	0.186 ^d^

Note: ^a^ one-way repeated measures ANOVA; ^b^ Friedman test; ^c^ partial eta squared; ^d^ Kendall’s W; MF—mental fatigue; ME—mental effort; MS—mental stress.

**Table 4 brainsci-15-00541-t004:** Results of behavioral differences between pre- and post-test in PVT.

Indicator	Pre-Test	Post-Test	*p*	Effect Size
	Mean ± SD	Mean ± SD		
Reaction Time (ms)	276.32 ± 27.22	291.29 ± 37.67	<0.001	−0.682
Attention Lapse	0.15 ± 0.43	0.54 ± 1.01	0.001	−0.754

Note: Wilcoxon signed-rank test.

## Data Availability

The data presented in this study are available on request from the corresponding author. The data are not publicly available due to privacy or ethical restrictions.

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
