# Peer review of "A New Method for Inducing Mental Fatigue: A High Mental Workload Task Paradigm Based on Complex Cognitive Abilities and Time Pressure"

_brainsci, 2025, doi:10.3390/brainsci15060541_

Round 1

Reviewer 1 Report

Comments and Suggestions for Authors

Reviewer’s report

The manuscript introduces a novel method for inducing mental fatigue (MF) using a high mental workload (HMW) task paradigm. The authors developed a 1-back Stroop (BS) task that integrates complex cognitive abilities and time pressure. They validated the effectiveness of this paradigm by comparing it to a traditional 2-back task and by assessing the impact of time-on-task (ToT). The study used subjective measures (NASA-TLX, VAS) and objective behavioral metrics (reaction time, accuracy) to evaluate MF induction.

Objective and Relevance

The objective of the paper is to develop and validate a new, more effective method for inducing MF. This objective is highly relevant. As the paper notes, mental fatigue is a significant concern in modern society, affecting psychological and physiological well-being, as well as task performance. The paper addresses limitations in existing MF induction methods and offers a potentially valuable tool for future research.

Here are some suggestions to improve the manuscript:

  1. I suggest to expand on the theoretical background of mental fatigue. While the authors mention several theories, a more in-depth discussion of resource depletion theory and its neural correlates would strengthen the introduction. Consider discussing the role of specific brain regions (e.g., prefrontal cortex, anterior cingulate cortex) in mental fatigue. Relevant literature includes for example: Langner, R., & Eickhoff, S. B. (2013). Sustaining attention to simple tasks: a meta-analytic review of the neural mechanisms of vigilant attention. Psychological bulletin, 139(4), 870–900. https://doi.org/10.1037/a0030694.
  2. In the methods section, please provide more detail about the BS task paradigm. The supplementary materials should be thoroughly described within the main text. Clarify how the time pressure was manipulated and why the specific time limits for each condition were chosen. A more detailed explanation of the task's cognitive demands would also be beneficial.
  3. In the results please include effect sizes (e.g., Cohen's d, partial eta-squared) for the statistical analyses. This would give readers a better understanding of the magnitude of the observed effects. Additionally, consider including correlation analyses to examine the relationships between subjective and objective measures of mental fatigue.
  4. Discuss the limitations of the study. For instance, the choice of a sample of young, male cadets from a military university might limit the generalizability of the findings. Also, discuss potential confounding factors, such as motivation and individual differences in cognitive abilities. Finally, mental fatigue has a very strong basis in neurophysiology (e.g., EEG, pupil dilation) while the authors used only subjective and behavioral measures. The discussion of potential other ways to measure MF could be discussed

By addressing these points, I believe the authors can strengthen their manuscript and increase its impact on the field.

Author Response

Submission ID: brainsci-3566445

Title: A New Method for Inducing Mental Fatigue: A High Mental Workload Task Paradigm Based on Complex Cognitive Abilities and Time Pressure

Dear Editor and Reviewers,

Appreciate for your attention and comments on our manuscript “A New Method for Inducing Mental Fatigue: A High Mental Workload Task Paradigm Based on Complex Cognitive Abilities and Time Pressure” (Submission ID brainsci-3566445). We have revised the manuscript according to your detailed advice and kind suggestions point-by-point and highlighted (by red color) all the amends on our revised manuscript.

Please find enclosed our revised manuscript and the detailed responses to the reviewers. We sincerely hope this manuscript will be finally acceptable to be published on Brain Sciences. Thank you very much for all your help and looking forward to hearing from you soon.

With best regards,

Yours sincerely,

Xufeng Liu, lxf_fmmu@163.com; Lin Wu, 1173843313@qq.com

Review 1

Comments and Suggestions for Authors

The manuscript introduces a novel method for inducing mental fatigue (MF) using a high mental workload (HMW) task paradigm. The authors developed a 1-back Stroop (BS) task that integrates complex cognitive abilities and time pressure. They validated the effectiveness of this paradigm by comparing it to a traditional 2-back task and by assessing the impact of time-on-task (ToT). The study used subjective measures (NASA-TLX, VAS) and objective behavioral metrics (reaction time, accuracy) to evaluate MF induction.

Objective and Relevance

The objective of the paper is to develop and validate a new, more effective method for inducing MF. This objective is highly relevant. As the paper notes, mental fatigue is a significant concern in modern society, affecting psychological and physiological well-being, as well as task performance. The paper addresses limitations in existing MF induction methods and offers a potentially valuable tool for future research.

Response:

Thank you for your kind compliments and constructive comments on our manuscript. We have done our best to revise the manuscript thoroughly and properly.

Here are some suggestions to improve the manuscript:

1.I suggest to expand on the theoretical background of mental fatigue. While the authors mention several theories, a more in-depth discussion of resource depletion theory and its neural correlates would strengthen the introduction. Consider discussing the role of specific brain regions (e.g., prefrontal cortex, anterior cingulate cortex) in mental fatigue. Relevant literature includes for example: Langner, R., & Eickhoff, S. B. (2013). Sustaining attention to simple tasks: a meta-analytic review of the neural mechanisms of vigilant attention. Psychological bulletin, 139(4), 870–900. https://doi.org/10.1037/a0030694.

Response:

We sincerely appreciate the reviewer’s constructive suggestions. In the revised manuscript, we have added some discussion of resource depletion theory and its neural correlates.

Please refer to Lines 61-73.

Thanks again for your comments.

2.In the methods section, please provide more detail about the BS task paradigm. The supplementary materials should be thoroughly described within the main text. Clarify how the time pressure was manipulated and why the specific time limits for each condition were chosen. A more detailed explanation of the task's cognitive demands would also be beneficial.

Response:

Thanks for your kind suggestions and we consider your comments very valuable.

As you pointed out, we have added supplementary materials to the main text, so that readers can clearly and completely understand the preparation process and key parameters of BS task. In addition, we also briefly explained the task’s cognitive demands in the methods section (Lines 124-126).

Please refer to Lines 124-158, 189-198, 272-293, and 314-335.

Thanks again for your comments.

3.In the results please include effect sizes (e.g., Cohen's d, partial eta-squared) for the statistical analyses. This would give readers a better understanding of the magnitude of the observed effects. Additionally, consider including correlation analyses to examine the relationships between subjective and objective measures of mental fatigue.

Response:

Thank you for your valuable suggestions to improve the interpretability of our results. We have added the effect sizes for the statistical analyses. Please refer to the tables (Table 2, Table 3 and Table 4) in the main text for specific details.

In addition, we have added content on correlation analysis.

Please refer to Lines 311-312 (Methods section); Lines 383-391 (Results section); Lines 485-494 (Discussion section).

Thanks again for your comments.

4.Discuss the limitations of the study. For instance, the choice of a sample of young, male cadets from a military university might limit the generalizability of the findings. Also, discuss potential confounding factors, such as motivation and individual differences in cognitive abilities. Finally, mental fatigue has a very strong basis in neurophysiology (e.g., EEG, pupil dilation) while the authors used only subjective and behavioral measures. The discussion of potential other ways to measure MF could be discussed.

Response:

Thanks for your kind suggestions and we consider your comments very valuable.

The particularity of the sample, the difference of individual cognitive ability and the lack of objective neurophysiological methods are also the limitations of this study, which we added in the limitation section.

Please refer to Lines 495-509.

Thanks again for your comments.

By addressing these points, I believe the authors can strengthen their manuscript and increase its impact on the field.

Response:

Thank you for your kind compliments and constructive comments on our manuscript. We have done our best to revise the manuscript thoroughly and properly.

We appreciate for Editors/Reviewers’ warm work earnestly, and hope that the correction will meet with approval. If you have any questions, please feel free to contact me. Thank you for your comments to make our research report more perfect.

Once again, thank you very much for your comments and suggestions.

With best regards,

Yours sincerely,

Xufeng Liu, lxf_fmmu@163.com; Lin Wu, 1173843313@qq.com

Reviewer 2 Report

Comments and Suggestions for Authors

The article is very interesting. I enjoyed reading it. At the beginning of the article, the authors review the scientific literature. It is quite comprehensive and covers the problem of cognitive fatigue from many angles.

I can recommend, for my part, to get acquainted with the concept of "payment for the mind".

For example, Madabhushi R., Gao F., Pfenning, A.R., Pan, L., Yamakawa, S., Seo, J., Rueda, R., Phan T. X., Yamakawa H., Pao P.C., Stott R.T., Gjoneska E., Nott A., Cho S., Kellis M., Tsai L.H. (2015). Activity-Induced DNA Breaks Govern the Expression of Neuronal Early-Response Genes. Cell, 161(7), 1592–1605. doi: 10.1016/j.cell.2015.05.032

I leave it to the author to decide whether to use it in their article. My recommendation is offered for review (I am not the author of the recommended article and I am not familiar with its authors).

The description of the participants needs to be clarified. I did not understand in the description of the participants what is meant by “average level of education — 14.99 ± 0.21 years” (p. 150). Is that how much they studied?

In the “results” the authors provide the values ​​of “behavioral indicators”. It is necessary to clarify what these indicators are.

- I recommend that after each table with the results the authors provide an explanation of all the abbreviations. This is necessary so that readers do not have to search for the explanation throughout the text.

The discussion of the results is done at the proper level. The authors gave an interpretation of the results and compared them with the results of other authors.

The conclusion summarizes the results of the study.

The list of references corresponds to the content of the manuscript.

Author Response

Submission ID: brainsci-3566445

Title: A New Method for Inducing Mental Fatigue: A High Mental Workload Task Paradigm Based on Complex Cognitive Abilities and Time Pressure

Dear Editor and Reviewers,

Appreciate for your attention and comments on our manuscript “A New Method for Inducing Mental Fatigue: A High Mental Workload Task Paradigm Based on Complex Cognitive Abilities and Time Pressure” (Submission ID brainsci-3566445). We have revised the manuscript according to your detailed advice and kind suggestions point-by-point and highlighted (by red color) all the amends on our revised manuscript.

Please find enclosed our revised manuscript and the detailed responses to the reviewers. We sincerely hope this manuscript will be finally acceptable to be published on Brain Sciences. Thank you very much for all your help and looking forward to hearing from you soon.

With best regards,

Yours sincerely,

Xufeng Liu, lxf_fmmu@163.com; Lin Wu, 1173843313@qq.com

Review 2

Comments and Suggestions for Authors

The article is very interesting. I enjoyed reading it. At the beginning of the article, the authors review the scientific literature. It is quite comprehensive and covers the problem of cognitive fatigue from many angles.

I can recommend, for my part, to get acquainted with the concept of "payment for the mind".

For example, Madabhushi R., Gao F., Pfenning, A.R., Pan, L., Yamakawa, S., Seo, J., Rueda, R., Phan T. X., Yamakawa H., Pao P.C., Stott R.T., Gjoneska E., Nott A., Cho S., Kellis M., Tsai L.H. (2015). Activity-Induced DNA Breaks Govern the Expression of Neuronal Early-Response Genes. Cell, 161(7), 1592–1605. doi: 10.1016/j.cell.2015.05.032

I leave it to the author to decide whether to use it in their article. My recommendation is offered for review (I am not the author of the recommended article and I am not familiar with its authors).

Response:

We sincerely appreciate the reviewer's positive feedback and thoughtful recommendation regarding the concept of “payment for the mind”. We carefully reviewed the suggested article by Madabhushi et al. (2015) and found its exploration of activity-induced DNA breaks in neuronal gene expression highly insightful. However, we were unable to locate a direct reference to the term “payment for the mind” within this work. To ensure thoroughness, we further searched PubMed and Web of Science using this keyword but did not identify closely related literature. Given the potential relevance of this concept to our discussion on mental fatigue, we would be grateful if the reviewer could provide additional clarification or references to guide our integration of this perspective.

Thanks again for your comments.

The description of the participants needs to be clarified. I did not understand in the description of the participants what is meant by “average level of education — 14.99 ± 0.21 years” (p. 150). Is that how much they studied?

Response:

Thanks for your kind suggestions and we consider your comments very valuable.

“average level of education — 14.99 ± 0.21 years” represents the total years of formal education completed by participants at the time of the study. Within the Chinese education system, formal education comprises: 5-6 years of primary school, 3 years of junior secondary school, 3 years of senior secondary school, 4-5 years of undergraduate education, and 3 years of postgraduate study, et al. Given that all participants in this study were medical university undergraduates, their reported education years fall within this expected range. We have explicitly clarified this measure as “Years of formal education” in the relevant section.

Please refer to Line 198.

Thanks again for your comments.

In the “results” the authors provide the values of “behavioral indicators”. It is necessary to clarify what these indicators are.

Response:

Thanks for your kind suggestions and we consider your comments very valuable.

In the validation of HMW induction effectiveness, behavioral indicators refers to Reaction Time and Accuracy in both the BS cognitive task and 2-back task, as presented in Table 2 (last two lines) and Figure 5 (ordinate axis).

In the validation of MF induction effectiveness, behavioral indicators refers to the count of Correct trials and Accuracy in the BS cognitive task, as presented in Table 3 (last two lines) and Figure 7 (ordinate axis), and Reaction Time and Attention Lapse in the PVT Task, as presented in Table 4 and Figure 9 (ordinate axis), respectively.

We have also made explanations at the corresponding positions.

Please refer to Lines 352-353, 374 and 393.

Thanks again for your comments.

- I recommend that after each table with the results the authors provide an explanation of all the abbreviations. This is necessary so that readers do not have to search for the explanation throughout the text.

Response:

Thanks for your kind suggestions and we consider your comments very valuable.

We have fully marked the abbreviations in the note of each table and figure so that readers can understand them quickly. Please refer to the related tables and figures in the main text for specific details.

Thanks again for your comments.

The discussion of the results is done at the proper level. The authors gave an interpretation of the results and compared them with the results of other authors.

The conclusion summarizes the results of the study.

The list of references corresponds to the content of the manuscript.

Response:

Thank you for your kind compliments and constructive comments on our manuscript. We have done our best to revise the manuscript thoroughly and properly.

We appreciate for Editors/Reviewers’ warm work earnestly, and hope that the correction will meet with approval. If you have any questions, please feel free to contact me. Thank you for your comments to make our research report more perfect.

Once again, thank you very much for your comments and suggestions.

With best regards,

Yours sincerely,

Xufeng Liu, lxf_fmmu@163.com; Lin Wu, 1173843313@qq.com

Reviewer 3 Report

Comments and Suggestions for Authors

Dear authors,

the submitted paper aims to propose a new method to induce fatigue in laboratory settings. The topic is quite relevant. As you pointed out, methods to induce fatigue vary widely and the majority of them are based on the assumption that time-on-task is the feature which most determines fatigue levels.

The paper is very clear and easy to read. The structure helps the readers to understand the concepts and the different steps of your research. Also the way in you present data, and discuss your findings is very easy to follow and that is always appreciated when reviewing/reading a paper.
I have no comments regarding the core of your work, while I would improve some minor aspect of the manuscript. Below a point-by.point list of what I would implement:

1) Cognirive flexibility, at line 111 -> some of the readers might not be familiar with this concept, as they could be with other concepts like mental effort, mental stress, and others. Up to you if you want to provide a short description of this concept.

2) VAS at line 130 -> Authors state that 4 VAS are provided along the experiment. It would be beneficial for the readers that even a short description of the vas or their item it is described in the main part of the manuscript

3) Line 142. Why participants are only males?And why 18-23 yo? I might assume that this was due to the demographic characteristic of students pool available when recruiting. However, this is not clearly states and readers might ask themselves if this was a deliberate choice or a contingency you had to adapt to...
Also, why participants of first step is described in supplementary materials and the remaining two are describe din the main body of the paper?

4) table 1 -> matched test is student t-test? If yes, I would use consistent terminology along the manuscript (see statistical analysis section)

5) figures -> I would add a title to the figures in the upper part so that is clearly and immediately understandable. As it is now, readers must read caption or y-axes label in order to get the focus of a fig. This would help readers especially when navigating through sections.

6) AT, EEG, HRV, at lines 376-377-> is there the need for acronyms? Since it appears that these are only discussed in this section, I think acronyms could be avoided.

These I listed are minor aspects of the paper, which I think is solid. I highlighted what I think could be improved, but the paper is of good quality already now.

Hope my comments will be of any help.
Bests.

Author Response

Submission ID: brainsci-3566445

Title: A New Method for Inducing Mental Fatigue: A High Mental Workload Task Paradigm Based on Complex Cognitive Abilities and Time Pressure

Dear Editor and Reviewers,

Appreciate for your attention and comments on our manuscript “A New Method for Inducing Mental Fatigue: A High Mental Workload Task Paradigm Based on Complex Cognitive Abilities and Time Pressure” (Submission ID brainsci-3566445). We have revised the manuscript according to your detailed advice and kind suggestions point-by-point and highlighted (by red color) all the amends on our revised manuscript.

Please find enclosed our revised manuscript and the detailed responses to the reviewers. We sincerely hope this manuscript will be finally acceptable to be published on Brain Sciences. Thank you very much for all your help and looking forward to hearing from you soon.

With best regards,

Yours sincerely,

Xufeng Liu, lxf_fmmu@163.com; Lin Wu, 1173843313@qq.com

Review 3

Comments and Suggestions for Authors

Dear authors,

the submitted paper aims to propose a new method to induce fatigue in laboratory settings. The topic is quite relevant. As you pointed out, methods to induce fatigue vary widely and the majority of them are based on the assumption that time-on-task is the feature which most determines fatigue levels.

The paper is very clear and easy to read. The structure helps the readers to understand the concepts and the different steps of your research. Also the way in you present data, and discuss your findings is very easy to follow and that is always appreciated when reviewing/reading a paper.

Response:

Thank you for your kind compliments and constructive comments on our manuscript. We have done our best to revise the manuscript thoroughly and properly.

I have no comments regarding the core of your work, while I would improve some minor aspect of the manuscript. Below a point-by-point list of what I would implement:

1) Cognitive flexibility, at line 111 -> some of the readers might not be familiar with this concept, as they could be with other concepts like mental effort, mental stress, and others. Up to you if you want to provide a short description of this concept.

Response:

Thanks for your kind suggestions and we consider your comments very valuable.

Cognitive flexibility refers to an individual’s capacity to adaptively modify thinking patterns, behavioral strategies, and attentional focus in response to changing task demands, environmental contingencies, or novel information. As a core component of advanced executive functions, it reflects the brain’s dynamic adaptive capabilities. In the present study, we operationalized cognitive flexibility as participants’ task-switching ability between the spatial 1-back and Stroop tasks.

Since we have incorporated the task complexity and time pressure construction procedures from the supplementary materials into the main manuscript, we have also provided a concise operational definition of “cognitive flexibility” in the Methods section.

In addition, the concepts of mental effort, mental stress, and boredom are discussed in the discussion section.

Please refer to Lines 134-135 and 443-454.

Thanks again for your comments.

2) VAS at line 130 -> Authors state that 4 VAS are provided along the experiment. It would be beneficial for the readers that even a short description of the vas or their item it is described in the main part of the manuscript

Response:

Thanks for your kind suggestions and we consider your comments very valuable.

In fact, we have described VAS in detail in section 2.3.5 of materials. In order to facilitate readers’ understanding, we also added a brief description of VAS and its four components at the corresponding position you mentioned.

Please refer to Lines 177-178 and 261-270.

Thanks again for your comments.

3) Line 142. Why participants are only males?And why 18-23 yo? I might assume that this was due to the demographic characteristic of students pool available when recruiting. However, this is not clearly states and readers might ask themselves if this was a deliberate choice or a contingency you had to adapt to...
Also, why participants of first step is described in supplementary materials and the remaining two are describe din the main body of the paper?

Response:

Thanks for your kind suggestions and we consider your comments very valuable.

Due to the particularity of military universities and the requirements of enrollment policy, the number of male students is far larger than that of female students, and male subjects are easier to obtain.
In the context of military tasks, compared with women, men are more likely to become mentally fatigued workers and have more job adaptability.

At the same time, in the context of China’s education, the length of schooling for most medical majors such as clinical medicine is 5 years, so the age range of 18-23 can basically cover all undergraduate students in medical universities.

However, the recruitment of subjects in this study is still limited. In the future, we should include women and expand the age range and the population background for in-depth research. The limitations of the subject population have been declared in the limitation section. We added the supplementary materials to the main text, so that readers can clearly and completely understand the preparation process and key parameters of the BS cognitive task.

Please refer to Lines 495-509.

Thanks again for your comments.

4) table 1 -> matched test is student t-test? If yes, I would use consistent terminology along the manuscript (see statistical analysis section)

Response:

Thanks for your kind suggestions and we consider your comments very valuable.

Generally speaking, student t-test includes one sample t-test, matched t-test and two sample t-test. Matched t-test is a kind of Student t-test. We have unified the terminology in the statistical analysis section. We apologize for the confusion caused by the failure to accurately express the statistical methods.

Please refer to Line 310.

Thanks again for your comments.

5) figures -> I would add a title to the figures in the upper part so that is clearly and immediately understandable. As it is now, readers must read caption or y-axes label in order to get the focus of a fig. This would help readers especially when navigating through sections.

Response:

Thank you for your constructive suggestion. We have revised all figures by adding concise descriptive titles at the top of each figure to immediately convey their key focus. We believe these changes enhance readability, especially when navigating through sections, and appreciate your guidance in improving the clarity of our presentation.

Thanks again for your comments.

6) AT, EEG, HRV, at lines 376-377-> is there the need for acronyms? Since it appears that these are only discussed in this section, I think acronyms could be avoided.

Response:

Thanks for your kind suggestions and we consider your comments very valuable.

After we incorporate the supplementary materials into the manuscript, we first mentioned available time (AT) in Section 2.1, so we used the abbreviation AT in the limitations section. The abbreviations of EEG and HRV have been deleted.

Thanks again for your comments.

These I listed are minor aspects of the paper, which I think is solid. I highlighted what I think could be improved, but the paper is of good quality already now.

Hope my comments will be of any help.
Bests.

Response:

Thank you for your kind compliments and constructive comments on our manuscript. We have done our best to revise the manuscript thoroughly and properly.

We appreciate for Editors/Reviewers’ warm work earnestly, and hope that the correction will meet with approval. If you have any questions, please feel free to contact me. Thank you for your comments to make our research report more perfect.

Once again, thank you very much for your comments and suggestions.

With best regards,

Yours sincerely,

Xufeng Liu, lxf_fmmu@163.com; Lin Wu, 1173843313@qq.com

Round 2

Reviewer 1 Report

Comments and Suggestions for Authors

Authors have addressed my concerns.